# Inactivation of Human Coronavirus by Titania Nanoparticle Coatings and UVC Radiation: Throwing Light on SARS-CoV-2

**DOI:** 10.3390/v13010019

**Published:** 2020-12-24

**Authors:** Svetlana Khaiboullina, Timsy Uppal, Nikhil Dhabarde, Vaidyanathan Ravi Subramanian, Subhash C. Verma

**Affiliations:** 1Department of Microbiology and Immunology, Reno School of Medicine, University of Nevada, 1664 N Virginia Street, Reno, NV 89557, USA; skhaiboullina@med.unr.edu (S.K.); tuppal@med.unr.edu (T.U.); 2Chemical and Materials Engineering Department, University of Nevada, LME 309, MS 388, Reno, NV 89557, USA; ndhabarde@nevada.unr.edu; 3GenNEXT Materials and Technologies, LLC., Reno, NV 89511, USA

**Keywords:** SARS-CoV-2, COVID-19, virus inactivation, UV, TNP coating

## Abstract

The newly identified pathogenic human coronavirus, SARS-CoV-2, led to an atypical pneumonia-like severe acute respiratory syndrome (SARS) outbreak called coronavirus disease 2019 (abbreviated as COVID-19). Currently, nearly 77 million cases have been confirmed worldwide with the highest numbers of COVID-19 cases in the United States. Individuals are getting vaccinated with recently approved vaccines, which are highly protective in suppressing COVID-19 symptoms but there will be a long way before the majority of individuals get vaccinated. In the meantime, safety precautions and effective disease control strategies appear to be vital for preventing the virus spread in public places. Due to the longevity of the virus on smooth surfaces, photocatalytic properties of “self-disinfecting/cleaning” surfaces appear to be a promising tool to help guide disinfection policies for controlling SARS-CoV-2 spread in high-traffic areas such as hospitals, grocery stores, airports, schools, and stadiums. Here, we explored the photocatalytic properties of nanosized TiO_2_ (TNPs) as induced by the UV radiation, towards virus deactivation. Our preliminary results using a close genetic relative of SAR-CoV-2, HCoV-NL63, showed the virucidal efficacy of photoactive TNPs deposited on glass coverslips, as examined by quantitative RT-qPCR and virus infectivity assays. Efforts to extrapolate the underlying concepts described in this study to SARS-CoV-2 are currently underway.

## 1. Introduction

The novel coronavirus, 2019-nCoV, SARS-CoV-2 causative agent of the pneumonia-like COVID-19 pandemic, has infected more than 77 million people globally [1]. Due to extremely transmissible nature of SARS-CoV-2 and limited available intensive care unit (ICU) management, over 1.7 million people have lost their life to this fatal SARS-CoV-2 outbreak, worldwide [2]. The highest rate of mortality is found among individuals aged 65 and over [3], whereas the death rate continues to decrease, but is still a concern among persons aged 85 years and over [4,5]. Devastating consequences of SARS-CoV-2 infection and ample amounts of vaccine or effective antivirals against SARS-CoV-2, warrants scientific research on this newly identified coronavirus for developing therapeutics and interventions to control viral spread. 

One of the main reasons for the observed surge in COVID-19 cases could be attributed to multiple modes of viral dissemination, including direct contact with infected person through respiratory droplets generated during coughing, sneezing, or talking, and indirect contact with contaminated surfaces or objects used by infected persons [6,7]. Analysis of 1070 specimens from multiple sites of 205 COVID-19-infected patients revealed the presence of viral load in urine, nasal, blood, sputum, and fecal samples [8]. Additionally, stability of SARS-CoV-2 in the environment also contributes to its spread [7]. Supporting the assumption of non-aerosol virus transmission are documented among COVID-19 cases, which could be explained by direct contact [9]. Due to the risk of nosocomial spread, clinical management of critically ill patients and infection prevention among health-care workers and non-COVID-19-infected patients requires the continuous use of personal protection and decontamination practices [10]. In addition, World Health Organization (WHO), recommends that maintaining physical distancing, frequent hand-washing and cleaning of the contaminated surfaces reduces the chance of contracting SARS-CoV-2 infection [11]. 

COVID-19 pandemic brought the importance of environmental hygiene and cleaning practices to prevent viral spread. COVID-19 virus was detected on multiple surfaces in the near proximity to patient’s bed as well as in the sink and the toilet bowl [12]. Investigation of the hospital cluster of cases identified respiratory droplets as the main source of the significant environmental contamination [13]. In another study, transmission of the virus via fomites (e.g., elevator buttons and public restrooms) was suggested in the cluster of COVID-19 cases in a mall in Wenzhou, China [14] as analyzed by RT-qPCR. However, detection of viral RNA in the environmental samples based on PCR-based arrays may not always indicate the presence of viable virus that could be transmitted. It is especially important to evaluate the pathogen survival in the environment as the COVID-19 virus was shown to retain the infectivity for up to 72 h on the stainless steel and plastic surfaces, commonly used in the medical settings [7]. 

The best way to control the spread of the COVID-19 pandemic is to reduce the viral spread. Strategies to control the spread include use of hand-sanitizers, facemasks, physical distancing, self-isolation of patients, regular cleaning of shared surfaces, and frequent use of disinfectants. Since fomites are one of the ways of virus transmission, it is crucial to identify methods that could reduce virus survival on the surfaces. In this regard, generation of self-disinfecting surfaces based on the advanced oxidation processes/photocatalysis initiated by ultraviolet (UV) radiation will be of great importance. UV irradiation-based methods are powerful disinfectants due to its germicidal ability and have shown effectiveness in reducing infections caused by pathogens. Additionally, nanosized titanium dioxide (TiO_2_-nanoparticles or TNPs) have been proven to act as effective photocatalysts and shown to have both bactericidal and virucidal properties [15]. The anti-viral effect of TiO_2_ has been demonstrated against influenza virus [16], which is transmitted via aerosol and causes respiratory tract infection, similar to COVID-19 [17]. However, the anti-viral efficacy of TiO_2_ against human coronavirus remains largely unknown. 

In this study, we sought to determine the disinfection ability of TNPs using an in vitro approach. Our data demonstrate that the photocatalytic properties of TNPs are highly effective in inactivating human coronavirus, HCoV-NL63 by reducing the viral genomic RNA stability and virus infectivity. HCoV-NL63 is an alpha coronavirus, which causes acute respiratory distress symptoms (ARDS) among infected individuals. This virus is a Bio Safety Level 2 pathogen and has high similarity [18] with SARS-CoV-2, therefore was used as a surrogate to determine the efficacy of TNPs in virus disinfection [19]. Our results demonstrated that TNP coatings drastically increase the efficacy of HCoV-NL63 inactivation and even a very brief exposure of UV was able to effectively eliminate infectious virus as well as viral genomic RNA. Importantly, the efficacy of TNP coatings, in inactivating the virus, was retained at multiple humid environments, suggesting this to be an effective measure for providing clean/disinfected surfaces in public and hospital settings. In conclusion, this study provides conclusive evidence that surfaces coated with TNPs, a non-toxic thin layer applied as paint, can enhance surface disinfection from human coronaviruses.

## 2. Materials and Methods 

### 2.1. Cells

Vero E6 and HEK293L cells (ATCC, Manassas, VA, USA) were maintained in Dulbecco’s modified Eagle medium (DMEM, Hyclone, Logan, UT, USA) supplemented with 10% fetal bovine serum (FBS, Atlanta Biologicals, now R&D Systems, Minneapolis, MN, USA), 2 mM L-glutamine, 25 U/mL penicillin, and 25 μg/mL streptomycin. Cells were grown at 37 °C in a humidified chamber supplemented with 5% CO_2_.

### 2.2. Human Coronavirus 

HCoV-NL63 strain is a human coronavirus and belongs to the family *Coronaviridae*, genus alphacoronavirus. HCoV-NL63 was obtained from BEI Resources (1.6 × 10^6^ TCID_50_/mL; lot# 70033870, NIAID, NIH; Manassas, VA, USA) and propagated in Vero cells by infecting the Vero cell monolayer with HCoV-NL63 for 2 h. Unattached virus was removed by washing followed by addition of fresh medium. Supernatant containing virus was harvested after 7 days, cell debris were removed by centrifugation before storing at −80 °C. TCID50 of the harvested virus was determined indirectly by RT-qPCR using a standard curve generated as described later (*2.6. RNA extraction and qPCR*). By this method the harvested virus showed to have 2.1 × 10^6^ TCID50/mL. An aliquot (100 µL) of this supernatant (HCoV-NL63) was used for extracting RNA and Reverse Transcriptase qPCR (RT-qPCR) for quantifying viral copies. All the assays were conducted under biosafety level 2+ (BSL-2+) containment.

### 2.3. TiO_2_ Nanoparticles (TNPs)-Coated Glass Coverslips

TiO_2_ was a generous gift from Degussa Corporation (Degussa Corp., Piscataway, NJ, USA). TNPs suspension (300 ug/mL) was prepared in deionized (DI) water and vortexed. An aliquot (600 μL) of TNPs was placed on the clean UV-treated glass coverslip and dried at 60 °C, which resulted into a semi-transparent coating. TNPs-coated coverslips were stored at room temperature (RT) until use.

### 2.4. UV Photocatalysis

An aliquot (100 μL, 2.1 × 10^5^ TCID_50_) of HCoV-NL63 virus was placed on TNPs-coated and uncoated coverslips (18 mm diameter, 1017.88 mm^2^) and exposed to the USHIO Germicidal Lamp (model G30T8; Cypress, CA, USA), which generates UV-C light (wavelength: 254 nm, 99 V, 30 W, 0.355 A) for various time points. The UV light source was placed 76 cm, 50 cm, and 10 cm from the bottom of the wells containing coverslips, which applied, calculated 2900 µW/cm^2^, 4300 µW/cm^2^, and 13,000 µW/cm^2^ energies, respectively (where µW = 10^−6^ J/s). Virus inactivation was also analyzed under different humid environments, 45%, 65%, and 85% relative humidities (Rh), by applying a known amount of virus on TNPs-coated and uncoated coverslips under indicated humidies. The humidity conditions were generated by using humidifier placed inside the biological safety cabinet (BSL2) and measuring the humidities with AcuRite Indoor Thermometer and Hygrometer with Humidity Gauge (Acurite, Inc., Nine Mile Falls, WA, USA). Humidity range (45–85%) was selected based on the average (45%) standard in-door humidity recommendation by the Center for Disease Control and Prevention (CDC) and the outdoor humidity registered in many parts of the world including the coastal areas, which have humidities as high as 85% [20,21]. Additionally, the efficacy of virus inactivation present as a dried droplet on TNP surface was evaluated by drying the virus (HCoV-NL63) as small liquid droplets before exposing to UV light. Since virus was present in the complete medium including FBS, protein content of the medium provided similar proteinaceous material as in the saliva or nasal secretions. For drying the liquid droplets, 10 small aliquots (10 μL each) for a total of 100 μL, were deposited on either TNPs-coated or uncoated coverslips and allowed to visibly dry (20 min, room temperature, 45% humidity) before exposing to the UV light. Same volume of liquid droplets, from the same virus stock, were deposited onto coverslips or TNPs-coated surfaces and exposed to the UV light for 30 s or 1 min without drying the droplets. Viruses from these teratments were recovered by adding 100 μL of PBS on the coverslips and allowing it to dissolve for 30 min at 37 °C. Total RNA was extracted by directly adding Trizol reagent onto the surfaces with dissolved viruses to ensure efficient collection of the viral genomic RNA. For determining the residual infectious virus, dissolved content was collected and applied onto a permissive, human embryonic kidney (HEK293L) cell monolayer for the infectivity assay. To ensure that all the residual virus was recovered from the control and TNP-coated surfaces, 100 μL PBS was added and incubated at 37 °C for 30 min for three subsequent times followed by combining them before adding onto the target cells. 

### 2.5. Virus Infectivity and Immunofluorescence Assay (IFA)

Control or the UV-treated viruses were added onto the human embryonic kidney cells for 2 h (37 °C, 5% CO_2_) before washing the dead and disintegrated virus and adding fresh medium. These cells were incubated for 48 h at 37 °C in a humidified chamber supplemented with 5% CO_2_. Since an infectious virus enters the cells to replicate and produce viral genomic RNA and proteins, detection of intracellular viral RNA and viral proteins confirms the presence of live virus. Infected cells were harvested for the extraction of total RNA and the detection of intracellular viral genomic RNA through RT-qPCR (*2.6. RNA extractions and qPCR*). Viral proteins were detected through immunofluorescence assay in which infected cells were fixed using 3:1, methanol: acetone and stored at −80 °C until used. Cells were permeabilized with 0.1% Triton X-100 for 30 min, washed (3×), and blocked (3% normal donkey serum, 0.5% BSA) for 1 h at RT. Cell monolayers were washed again (3×) and incubated with polyclonal rabbit anti-CoV spike antibody (1:200; BEI Resources, NIAID, NIH) for 1 h at RT, followed by incubation with goat anti-rabbit Alexa Fluor 488 (1:5000; Molecular Probe, Carlsbad, CA, USA), secondary antibody for 1 h at RT in the dark. Finally, the nuclei were stained with TO PRO-3 (ThermoFisher, Walthman, MA, USA). Coverslips were mounted on the glass slides using antifade and the slides were examined using Carl Zeiss LSM 780 confocal laser-scanning microscope. 

### 2.6. RNA Extraction and qPCR

HEK293L cell monolayer infected with control or UV-treated virus were subjected for total RNA extraction extraction using Trizol reagent (Invitrogen, Carlsbad, CA, USA) by directly adding the reagent onto the cells for lysis according to the manufacturer’s recommendation. An aliquot of RNA (1 μg) was used for synthesizing the cDNA (Superscript kit; Invitrogen, Carlsbad, CA, USA). cDNA (2 μL) was used for the relative quantification of viral genome RNA in a qPCR assay (ThermoFisher Scientific, Waltham, MA, USA). Primers used in this study are NL63-SF 5′-GTGCCATGACCGCTGTTAAT-3′ and NL63-SR 5′-GCGGACGAACAGGAATCAAA-3′. For generating a standard curve to estimate the viral copies, 100 μL of NL63 (BEI resources, Manassas, VA, USA) stock with known amounts of HCoV-NL63 was used for total RNA extraction and cDNA synthesis. An aliquot (2 μL) of HCoV-NL63 cDNA used for RT-qPCR was equivalent to 320 copies of HCoV-NL63 (1.6 × 10^6^ TCID_50_/mL; lot# 70033870, NIAID, NIH). Several 10-fold dilutions of this isolated genomic RNA (BEI Resources, Manassas, VA, USA) were used for generating a standard curve. The sensitivity of qPCR assay was 3.2 virions in the sample based on the quantitation of RNA present in HCoV-NL63 virus stock from the BEI Resources.

### 2.7. Experimental Setup 

In order to test the efficacy of TNPs in accelerating the inactivation of HCoV-NL63 upon exposure to UV-light, a suspension of TNPs was prepared and deposited on glass coverslips as a semitransparent film. TNP-coated coverslips were then placed in the center of a 60 mm culture dish and an aliquot of HCoV-NL63 virus was applied on the top of coverslips, followed by exposure to UV emitting lamps for the selected time duration. After TNP/UV-treatment, the inactivated virus was collected and evaluated for the viral genomic RNA stability and virus infectivity. The experimental plan depicting these steps is presented as a graphical image (Figure 1).

### 2.8. Statistical Analysis

Data presented are an average of three independent experiments and the error bars represent standard deviation. Statistical analyses were performed using Prism 8.0 software (Graphpad Inc., La Jolla, CA, USA) and the *p*-values were calculated using 2-way ANOVA and the *p*-values are * <0.1, and ** <0.01.

## 3. Results

### 3.1. The Effect of UV Light Exposure on HCoV-NL63 Virus Stability

HCoV-NL63 virus (100 μL suspension) was applied onto the coverslip placed inside the wells of 12-well plate. The plate containing HCoV-NL63 viruses was exposed, after removing the lid, to the UV light inside a BSL-2 biological safety cabinet for 0, 1, 5, 10, and 30-min. UV-exposed virus was collected for the evaluation of virus inactivation by direct genomic RNA quantitation (RT-qPCR) and infectivity assay on HEK293L monolayer. Presence of the viral genomic RNA and its integrity was determined using qPCR as it was previously used for the detection of SARS-CoV-2 in environmental samples [13,22,23]. Genomic approaches are shown to be highly sensitive in detecting viral RNA and multiple targets are used for generating PCR amplicon [13,24]. Importantly, detection of genomic RNA through RT-qPCR does not constitute for the presence of an ‘infectious’ virus but shows the presence of viral genomic RNA. Therefore, a virus infectivity assay must be performed for the detection of any infectious virus. However, if there isn’t detection of intact viral genomic RNA, in samples after treatment/exposure, it can be concluded that virus has been completely eliminated. Any region of the viral genome can be used for detecting genome integrity and for our assays, we selected Spike protein, involved in attaching the virus to host cells [25], because of its high sensitivity and specificity for the detection of viral genomic RNA. HCoV-NL63 viral copies were calculated using standard curve generated on the basis of serial dilutions of known HCoV-NL63 genomic RNA. Expectedly, the copies of intact viral genomic RNA declined with UV-exposure (Figure 2A). Importantly, we saw an efficient reduction in the number of intact viral genomic RNA even at 1 min exposure to the UV-light. However, there were some intact genomic RNA of HCoV-NL63, at least in the region (spike protein), while other genomic regions may have been fragmented (to make the virus inactive) but increasing the exposure time to 5, 10, and 30 min completely degraded the genomic RNA of HCoV-NL63, below the detection limit, confirming total disintegration of the virus (Figure 2A). Since, RT-qPCR analyzes the viral genome, not the infectiousness, we asked whether treatment with UV can reduce the infectivity of HCoV-NL63 virus. To this end, UV exposed HCoV-NL63 viruses were added onto the monolayer of HEK293L for 2 h to facilitate attachment and entry of the residual infectious virus. The permissive nature of HEK293L cells were evaluated, before using them for the estimation of residual live virus, by adding same amounts of HCoV-NL63 on Vero E6 or HEK293L cells and allowing them to grow for 48 h after removing the unattached virus. Infection and replication of HCoV-NL63 virus in these cells were detected by localizing viral spike glycoprotein through Immunofluorescence assay. Detection of similar, if not better, amounts of viral protein staining in HEK293L cells confirmed them to be permissive for HCoV-NL63 infection. A representative image is shown in Figure 2C. Localization of viral proteins in the cells are used for infectivity assays but detection of intracellular viral genomic copies, following infection, can also be used for indirect quantitation of infectious virus besides gold standard assays (plaque assay). For our assay, total RNA collected at 48 hpi (hour post infection) was used for the detection of HCoV-NL63 genomic RNA, to measure the levels of residual live virus in UV exposed samples (Figure 2B). Our data showed that samples exposed to UV for 5-min or longer did not have any intracellular HCoV-NL63 genome suggesting total inactivation of the virus (Figure 2B). However, samples exposed to UV for 1min showed some levels of intracellular viral genomic RNA, suggesting few but very minimal levels of residual live virus, which entered and replicated in these cells. UV-untreated samples, which showed significant copies of intracellular genomic RNA (Figure 2B, 0 min), was used as a reference for estimating the levels of residual live virus.

### 3.2. TNP Coating Efficiently Inactivated HCoV-NL63 Even at Brief Exposures

Next, we sought to determine whether the photocatalytic activity of TNPs can facilitate/enhance the efficacy of virus inactivation. In order to do that, we exposed HCoV-NL63 virus with UV deposited on a surfaces coated with TNPs, which was achieved by depositing nanosized TNPs (600 μL) onto the coverslip (2.5 cm^2^ surface area) (Figure 1). HCoV-NL63 coronavirus (100 μL) was applied onto TNPs-coated and uncoated (control) coverslips, placed inside the wells of a 12-well plate and exposed to the UV light for indicated times. Total RNA was extracted by directly adding the Trizol onto the coverslip for quantifying the intact genomic HCoV-NL63 RNA copies through RT-qPCR, as described above. Expectedly, the viral copies quantified through RT-qPCR for viral genome fragmentation showed significantly reduced copies (almost to the background level) of HCoV-NL63 from the TNP-coated surface as compared to the control (uncoated surface) even at 1-min of UV-exposure (Figure 3A). 

We detected a slight variation in the number of recovered HCoV-NL63 copies from no UV treated samples (0 min UV-exposure), which could be because of the anti-microbial property of TNP even in the absence of UV exposure. Not surprisingly, the number of intact HCoV-NL63 genomic RNA was almost to the background levels at longer UV-exposures on both TNP and control surfaces (Figure 3A). To demonstrate whether the viral genomic RNA detected after the UV-treatment on the TNP or control surface had any infectious virus, a virus infectivity assays was performed by adding the recovered virus onto the monolayer of HEK293L cells. The residual infectious virus were determined by the localization of viral protein (spike) of HCoV-NL63 through immunofluorescence assay, which relies on the production of viral antigens following replication/transcription/translation. As detection of viral proteins in cells inoculated with UV-treated samples confirms the presence of infectious virus and the lack of any signal for the viral protein is indicative of viral inactivation. As a control of infection and viral replication, stock HCoV-NL63 was added onto HEK293L cells, stained for viral protein and the detection of proteins signals in HCoV-NL63 infected cells but not in uninfected control confirmed the specificity of this assay (Figure 3B, compare panel A with E). Cells infected with UV-treated samples showed that viruses on the TNPs-coated surfaces were completely inactivated even at 1 min exposure to UV-light, while control, surface had detectable levels of live infectious virus, demonstrated by the presence of immunofluorescent signal (Figure 3B, compare panels B and F). However, longer exposure to UV inactivated the virus to undetectable levels, demonstrated by the lack of immunofluorescent signals (Figure 3B, panels C, D, G, H). These data confirmed that virus on the TNPs-coated surface can be inactivated quickly as compared to the non-coated control surface.

### 3.3. TiO_2_ Coatings Effectively Inactivated HCoV-NL63 Virus Present in Wet or Dried Form

We wanted to determine whether TNP coating is effective in inactivating viruses that have been dried after falling onto the surfaces, to emulate the most common settings where respiratory/sneeze droplets coming out of the infected individuals are deposited on surfaces. To evaluate whether HCoV-NL63 virus fallen on the TNP-coated surface can be inactivated even after drying, we added a small droplets of medium containing a known amounts of HCoV-NL63 on TNP-coated surface as well as on the control surface, without TNP coating. These droplets were allowed to dry, which took approximately 20 min at room temperature and normal humidity (45% RH) to visibly dry, followed by exposing them to the UV light for indicated times (Figure 4). HCoV-NL63 virus on TNP-coated or uncoated surfaces as droplets, without drying, was used as a control to evaluate the effects of drying on viral inactivation. HCoV-NL63 viruses from both dried as well as liquid droplets were recovered following 0 min (control), 0.5 min, and 1min of UV exposure, by dissolving them in PBS, as mentioned in the method section. Quantification of HCoV-NL63 viral genome copies showed significantly reduced viral copies in dried samples, even without UV exposure, among both TNP-coated as well as control surfaces, suggesting that drying itself degraded viral genome to some extent (Figure 4A). UV exposure further fragmented the viral genome present in the liquid droplets as well as in the dried form but most importantly, viruses deposited on TNP-coated surface were degraded highly effectively when compared to the control surface (Figure 4A). These results confirmed that TNP-coated surfaces are highly efficient in degrading viral genome copies regardless of whether it is present in a liquid or dried form. We further tested the copies of infectious virus through infectivity assay. Recovered virus after UV treatment were added onto HEK293L cells for 48 h to enter into the cells and replicate. Estimation of intracellular viral copies through RT-qPCR confirmed that virus recovered from TNP surface did not have any infectious virus. Understandably, lower copies of the virus in dried samples may also have been due to an inefficient recovery of the virus, although we ensured to dissolve the dried droplet in PBS before extracting RNA, a significant reduction of viral copies in the dried samples confirmed that TNP-coated surfaces efficiently inactivates human coronavirus. 

### 3.4. Reducing the Exposure Distance and Increasing the Relative Humidity Enhanced HCoV-NL63 Inactivation

We further determined whether decreasing the distance of UV-source from the contaminated surface will reduce the exposure time for complete disinfection. To this end, we placed a known (calculated) amount of HCoV-NL63 virus on a normal or TNP-coated coverslips before exposing them to UV source from a distance of 10 cm or 50 cm, which provided a power of 13,000 µM/cm^2^ and 4300 µM/cm^2^, respectively. Expectedly, exposing the surface from a distance of 50 cm (4300 µM/cm^2^) disintegrated the virus genomic RNA more effectively with only a limited number of detectable viral genome at 1 min exposure. Importantly, TiO_2_ significantly decreased the exposure time to 0.5 min to achieve almost complete disintegration of the viral genome (Figure 5A). Further decreasing the distance to 10 cm disintegrated the viral genome very quickly to below the detection limit in less than 0.5 min on the control as well TiO_2_ surfaces (Figure 5A). 

We also determined the effects of relative humidity on viral genome degradation by incubating the surfaces containing virus under different relative humidities during UV exposure. We used three different relative humidities (45–85% Rh) to cover the range of humidities, from dry to humid environments, to UV treat the virus on TNP-coated or control surfaces. The surfaces with virus were exposed to the UV-light (76 cm distance with 2900 µW/cm^2^) for indicated times. Our results showed that an increase in the relative humidity enhanced HCoV-NL63 viral genome degradation with UV light on both control and TNP-coated surfaces (Figure 5B). Importantly, TNP coating enhanced the degradation of HCoV-NL63 virus at all three tested relative humidities (Figure 5B). These results confirmed that surface disinfection of viruses can be achieved more quickly on TNP-coated surface and the humidity can enhance the photocatalytic activity of TiO_2_.

## 4. Discussion

Highly contagious nature of SARS-CoV-2 highlights the need for environmental control of the viral spread. Stability of the virus in different environmental conditions [26] and the ability to spread far distances made this virus exceptionally successful in dissemination within the human population [27]. SARS-CoV-2 appears to be highly stable on smooth surfaces for up to 72 h and in a wide range of pH conditions (pH 3–10) at room temperature (22 °C) [27,28]. The virus can spread on surfaces by direct contact or aerosol droplets. The small-sized droplets (10 μm) can become airborne and circulate in the air for an extended period of time carrying the infectious virus to far distances [29]. They may be dispersed widely by air contaminating the environment far distant from the patient. Therefore, decontaminating surfaces is essential to combat COVID-19 spread.

Nanosized TiO_2_/TNPs have photocatalytic properties, which are shown to control microbial growth [15,30]. The mechanism of antimicrobial activity is based on the excitation of the electron/e^-^ from the valence band to the conduction band and the generation of the “electron-hole (e^−^h+)” [31]. The free electron can contribute to production of the reactive oxygen species (ROS) including O_2_^●^^‾^ and OH^●^ radicals [32]. These ROS can react in the solution producing highly potent anti-microbial H_2_O_2_. The TNPs can be used in suspension, liquids, or immobilized on surfaces, also referred to as a “self-cleaning“ surfaces [33]. This feature of TNPs could provide a microbial control in addition to conventional disinfecting products. TNP’s anti-viral effect has been shown against influenza virus, Newcastle virus, hepatitis B virus, and herpesviruses [34,35,36,37]. TiO_2_ was proposed to have an antiviral effect, targeting both DNA and RNA viruses, airborne and bloodborne pathogens. The photocatalytic titanium apatite filter was earlier shown to inactivate SARS coronavirus up to 99.99% after 6 h exposure [38]. Our assay to use TiO_2_ coating to efficiently inactivate human coronavirus with UV light through very brief exposure highlights the importance of TNP coatings on publicly used surfaces. Although, UV by itself is used for viral inactivation including the trains of public transport systems but the efficacies are not well established. Since our assays showed augmentation of virus inactivation, HCoV-NL63 by the TNPs coatings under UV exposure, we propose that TNP coatings can enhance virus inactivation including SARS-CoV-2 on surfaces most frequently exposed with COVID-19 patients. HCoV-NL63 and SARS-CoV-2 both cause acute respiratory diseases and have similar virion structure (HCoV) [18,39], and we speculate that these inactivation parameters will be same for the both viruses. 

Various factors are shown to affect virus resistance to UV light exposure, including UV dose, humidity, and the surface material [7,40]. It was reported that the virus survival decreases with an increase in the UV energy [41]. Recently, Buonnano et al. demonstrated that alpha and beta coronaviruses are susceptible to the low dose of the UV light exposure, where 90% of virus inactivation was achieved in 8 min [42]. The UV light susceptibility of SARS-CoV-2 was also demonstrated, confirming the virucidal effect of the UV light [43]. In both studies, the effect of the higher UV light intensity was shown to have more virucidal effect, as expected [42,43]. In line with these data, we have shown the beta coronavirus is susceptible to the UV radiation and that this effect is directly proportional to the UV-light intensity. The most interesting observation is that TiO_2_ can facilitate virucidal effect of the low intensity UV light, which could have important practical applications. As TiO_2_ can maintain antiviral efficacy of the UV light at the lower intensity, it will help in reducing the potential harmful effect of radiation exposure [44].

Interestingly, TiO_2_ nanoparticles effectively inactivated HCoV-NL63 under all three tested relative humidity conditions, which are within the standard for the indoor conditions (30–60%) [45]. We have shown that TiO_2_ nanoparticles retain virucidal efficacy even at very high humid condition (85% relative humidity), confirming a broader use of TNPs coatings on outdoor surfaces. The environmental humidity was suggested to play a role in the airborne virus spread [46,47,48]. It is postulated that humid environment contributes to the size of droplets and the stability of the virions [29,49]. A prolonged survival of the airborne viruses was demonstrated in the lower humid conditions [47,50]. A study by Wu et al. demonstrated a negative correlation between COVID-19 cases and humidity levels [51]. A study published by Matson et al. [52] showed that lower humidity combined with lower temperature prolonged the half-life of virus in nasal mucosa. Matson et al. also stated that virus is maintained in the climate controlled environment with relative humidity 40%, for 24 h [52]. It appears that environment factors can alter the survival of coronavirus on surfaces. Sizun et al. demonstrated that dried virus can still be infectious for hours on aluminum, surgical gloves, and sponges [53]. In addition, the dried virus was shown to retain the viability for over five days on smooth surfaces at air-conditioned environments [47]. Our data demonstrate that TiO_2_-coated surfaces have a viral inactivation property even on the virus that has been dried. Therefore, the in-vitro viral inactivation property of the TiO_2_ could be used for preventing the persistence of virus in dry droplets on surfaces.

A presumed model to understand the underlying mechanism involved in the photocatalytic inactivation of HCoV-NL63 has been described in Figure 6. In the presence of UV radiation, TiO_2_ nanoparticles act as photocatalyst due to their specific energy structure. UV-irradiation of nanosized TiO_2_ results in the excitation of electron leading to the formation of a specific “electron-hole (e^−^-h^+^)” pair and generation of reactive oxygen species (ROS). The electron holes (h^+^) catalyze oxidation processes and convert water/hydroxide molecules to peroxide/hydroxyl radicals, whereas electrons (e^−^) induce reduction reactions and react with molecular oxygen to generate superoxide radicals.

These resulting ROS have the ability to inactivate the virus through oxidative damage.

Since TiO_2_ is non-toxic and used as a food additives [54], although at low quantities, TNPs long-term effects on human health has not been evaluated. Therefore, a well-controlled toxicology studies along with approvals from the environmental protection agency and associated regulatory bodies may be needed, before using the TNPs on commonly used surfaces. TNPs have many advantages as compared to conventional germicidal products including its low cost and chemical stability [55]. Additionally, they do not require multiple applications as they could be coated single time on various surfaces [56,57]. In fairness, TNPs can be applied as a paint on the surfaces in the form of immobilized thin coatings on high risk facilities including train stations and other public areas to reduce the surface contamination and any possible exposures. It is an obvious advantage of TNPs that they have a broad range of anti-microbial efficacy including virucidal, bactericidal, and fungicidal [15]. This is especially beneficial in the areas with high risk of exposure to biohazardous material, such as hospitals. Additionally, this may be useful in hospital settings including the ICUs handling COVID-19 patients, where TNP coatings of the hospital floors can provide an effective way of inactivating viruses and surface contamination of SARS-CoV-2. In conclusion, we highlight the effectiveness of UV light-induced virucidal activity of TNPs in controlling coronavirus spread, which can help create a sanitary environment. 

In conclusion, the photoactive TNPs have strong virucidal effect, which are preserved under different environmental condition. In most cases, the genomic copies of human coronavirus NL63 from TNP-coated surface was undetectable within 1 min of UV exposure, which was also confirmed by cell infectivity assay. In an in-vitro virus inactivation assay, when the detection of viral RNA by the RT-qPCR is negative (or below the detection limit) it can be concluded that viral genome has been completely disintegrated. However, in the case of a positive RT-qPCR results, viral genome in the region of PCR amplification may be intact but fragmentation in other parts of the viral genome as well as inactivation of surface glycoprotein could make the virus non-infectious. Therefore, a virus infectivity assay must be performed to determine the levels of infectious virus. However, lack of a RT-qPCR signals can be construed as total inactivation of the virus though disintegration of viral genomic RNA.

## 5. Patents

The work has been submitted for provisional patent filling consideration. 

## Figures and Tables

**Figure 1 viruses-13-00019-f001:**
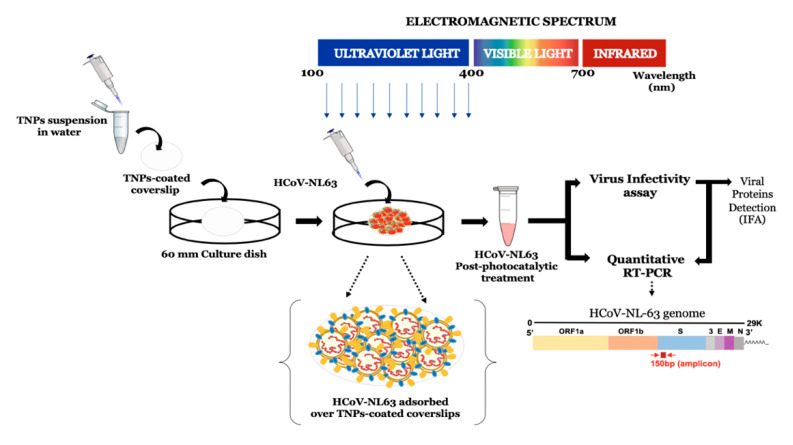
Schematic representation of the experimental set up utilized for the evaluation of photocatalytic inactivation of HCoV-NL63. TiO_2_ nanoparticles (TNPs) were prepared as a suspension in water and deposited on glass coverslips by drying at 60 °C for 4–5 h. An aliquot (100 µL) with calculated virions of HCoV-NL63 was placed on the TNP-coated coverslips and exposed to UV for indicated times. The TNP/UV-treated virus was then collected to determine the viral RNA stability and infectivity via quantitative RT-PCR, infectivity, and the detection of viral protein through immunofluorescence assay in HEK293L cells.

**Figure 2 viruses-13-00019-f002:**
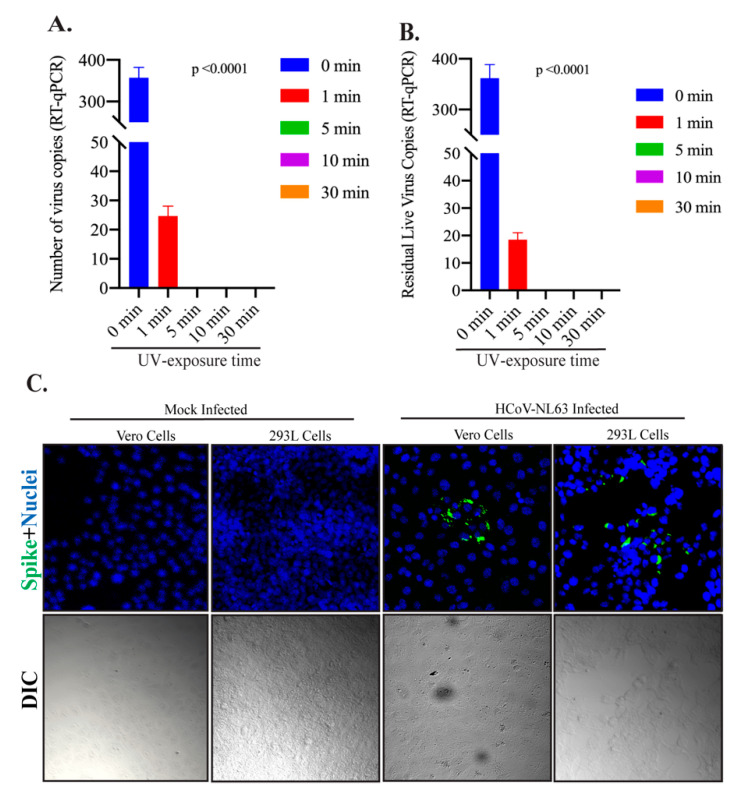
The effect of UV exposure on HCoV-NL63 viral genomic RNA stability. (**A**) HCoV-NL63 virus (100 µL) was placed on coverslips and exposed to UV light for indicated time (0, 1, 5, 10, and 30 min). Total RNA was extracted and used for the detection of viral copies by RT-qPCR. (**B**) HCoV-NL63 virus (100 µL) was exposed to UV light for indicated time (0, 1, 5, 10, and 30 min), collected, and used for the infection of HEK293L cells. Total RNA was extracted at 48 h post-infection for the detection of intracellular viral genomic copies in a RT-qPCR assay. Viral copies were calculated based on a standard curve generated using the known amounts of virus obtained from BEI Resources (material and method section). (**C**) 100 µL of HCoV-NL63 was added onto a monolayer of Vero E6 or HEK293L cells and incubated for 48 h before fixing them for immunofluorescence assay. HCoV-NL63 infected as well as respective mock infected cells were stained with anti-CoV antibody for spike glycoprotein. Immune localization of spike proteins showed specific signals (green fluorescent dots) in HCoV-NL63 infected but not in mock infected cells. Nuclei were stained with TO PRO-3 (blue signals). Differential Interference Contarst (DIC) images were captured to see the cellular morphology.

**Figure 3 viruses-13-00019-f003:**
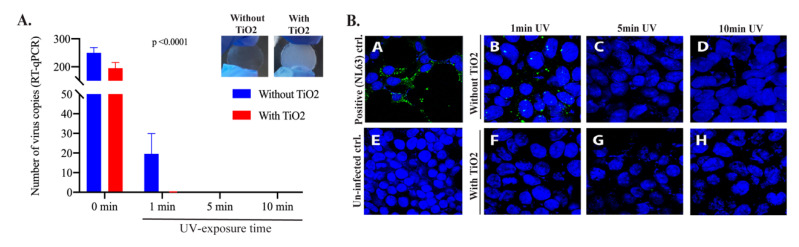
TiO_2_ nanoparticles enhanced HCoV-NL63 inactivation with no live virus in 1 min UV exposure. (**A**) HCoV-NL63 virus aliquot (100 µL) was placed on glass coverslips treated with or without TiO_2_ and exposed to UV light for 1, 5- and 10-min. Post-UV treatment, HCoV-NL63 was collected for viral RNA extraction. Intact viral copies were calculated based on the standard curve generated above. On top, coverslips without and with TNP coatings to show the level of opacity following TiO_2_ coatings. (**B**). Post-UV treatment, HCoV-NL63 virus was collected and subjected to the infection of HEK293L monolayer. Viral protein, indicator of virus infectivity and replication, was detected by immune localization through IFA (green fluorescent dots). A) NL63 infection (positive control); B) 1 min UV light exposure without TiO_2_; C) 5 min UV light exposure without TiO_2_; D) 10 min UV light exposure without TiO_2_; E) uninfected control (negative control); F) 1 min UV light exposure with TiO_2_; G) 5 min UV light exposure with TiO_2_; H) 10 min UV light exposure with TiO_2_.

**Figure 4 viruses-13-00019-f004:**
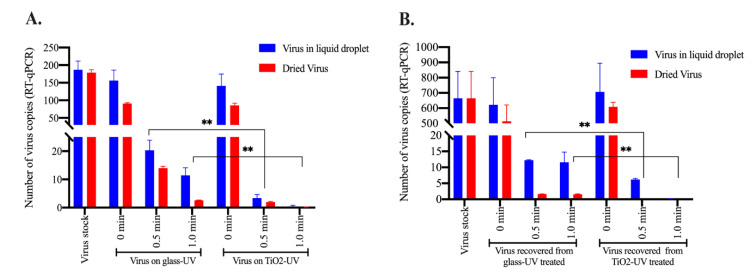
TNP-coated surface efficiently inactivated virus present in wet or dried form. HCoV-NL63 virus was deposited as 10µL droplets at 10 spots (100 µL total) on on glass coverslips treated with or without TiO_2_. One set was allowed to dry, which took ~20min to visibly dry before exposing to the UV light. Another set was exposed to the UV-light immediately after depositing the virus on control or the TNP surface. Virus copies were recovered by adding PBS. (**A**) Total RNA was extracted for the quantitation of intact viral genome copies through RT-qPCR. (**B**) Virus recovered in PBS was added onto a monolayer of HEK293L cells, allowed to grow for 48h after removing the unattached virus after 2 h. Total RNA from the infected cells were extracted for the quantitation of intracellular viral copies, following infection with residual infectious virus. Statistical analyses were performed using Prism 8.0 software and the *p*-values were calculated using 2-way ANOVA and the *p*-values are ** <0.01.

**Figure 5 viruses-13-00019-f005:**
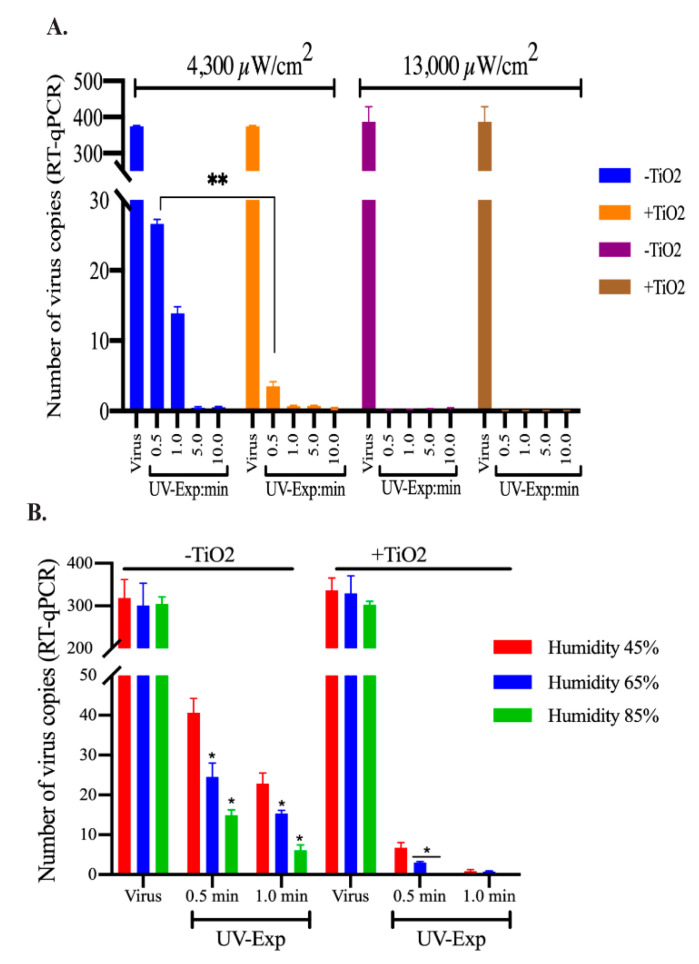
UV energy and humidity affected TNP mediated virus inactivation. (**A**) HCoV-NL63 virus aliquot (100 µL) was placed on glass coverslips treated with or without TiO_2_ and exposed to UV light for 0.5, 1, 5, and 10-min from a distance of 50 cm or 10 cm to achieve 4300 µM/cm^2^ and 13,000 µM/cm^2^ energy levels. Post-UV treatment, HCoV-NL63 was collected for viral RNA extraction. Intact viral copies were calculated based on the standard curve generated above. (**B**). HCoV-NL63 virus aliquot (100 µL) placed on TNP-coated or control (uncoated) coverslips were exposed to UV-lights for 0.5 and 1.0 min under 45%, 65%, and 85% relative humidities. Viruses from UV-untreated surfaces were used as controls. Post-UV treatment, viral genomic RNA was extracted and subjected for quantitation of genomic RNA in qPCR. Statistical analyses were performed using Prism 8.0 software and the *p*-values were calculated using 2-way ANOVA and the *p*-values are * <0.1, and ** <0.01.

**Figure 6 viruses-13-00019-f006:**
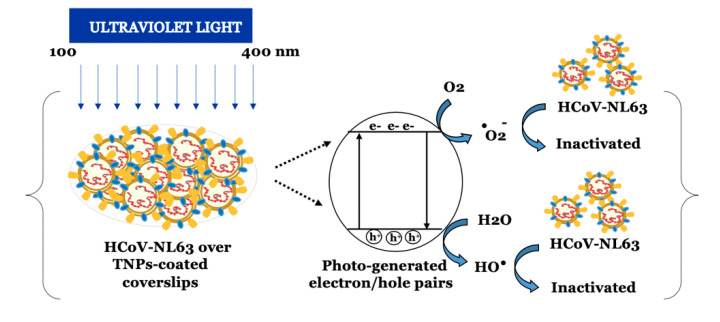
Schematic illustration of photocatalytic inactivation of HCoV-NL63 principle. Irradiation with UV light generates specific “electron-hole (e^−^-h^+^)” pair, and reactive oxygen species (ROS), including hydroxyl and superoxide radicals on the surface of TNPs. These ROS have strong oxidative ability, which can inactivate viruses through oxidative damage.

## Data Availability

The data presented in this study are available within this manuscript Khaiboullina et al. Viruses.

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
