# Peer review of "Inactivation of Human Coronavirus by Titania Nanoparticle Coatings and UVC Radiation: Throwing Light on SARS-CoV-2"

_viruses, 2020, doi:10.3390/v13010019_

Round 1

Reviewer 1 Report

This resubmitted version of the manuscript demonstrates significant improvements in quality and clarity compared to the previous version. The authors have clearly described their methods in detail, and explained the rationale for their approaches, particularly the relative humidity (RH) conditions used in their experiments. As public spaces continue to reopen, TNPs represent a promising approach to non-pharmaceutical interventions for reducing transmission of SARS-CoV-2 and other respiratory pathogens. Several minor comments are outlined below.

Comments:

  1. Mode of Virus Inactivation – The authors have set up their inactivation experiments by placing a 100 uL aliquot of virus on glass coverslips. Compared to respiratory droplets released from the human airway, this is a very large volume that is likely to remain in the liquid state for the duration of the assay (although this point should be clarified by the authors). Much smaller respiratory droplets and aerosols that may drive fomite transmission are likely to dry on high-touch surfaces. Droplet drying may act to stabilize the virus and may also affect the inactivation efficiently by UV/TNP. The authors demonstrated this effect themselves in Figure 4, where more viral genome copies were detected in drier conditions, regardless of inactivation approach. Which there is nothing inherently wrong with the authors’ approach in this manuscript, they must include the caveat in their Discussion (see also lines 345-346) that UV/TNP inactivation must also be analyzed in dried virus samples on various surfaces. The authors have also not assessed virus inactivation in a biologically relevant medium, which is another important caveat to be addressed here.
  2. Section 3.3 Title – The authors find that increasing RH accelerates virus inactivation (Figure 4). However, the title of this section indicates that reduced RH enhances HCoV-NL63 inactivation, which is not consistent with their findings.
  3. Missing reference – Reference 20 is incomplete (missing access date).
  4. Line 19 – “to combat COIVD-19 is” should be “to combat COVID-19 are”.
  5. Line 146 – “was used for” should be “were used for”.
  6. Line 253 – “amounts” should be singular.

Author Response

We thank you for your time and effort in reviewing our manuscript and providing insightful comments. We have addressed all the concerns in this revised manuscript and point by point response is appended below.

Comments:

Mode of Virus Inactivation – The authors have set up their inactivation experiments by placing a 100 uL aliquot of virus on glass coverslips. Compared to respiratory droplets released from the human airway, this is a very large volume that is likely to remain in the liquid state for the duration of the assay (although this point should be clarified by the authors). Much smaller respiratory droplets and aerosols that may drive fomite transmission are likely to dry on high-touch surfaces. Droplet drying may act to stabilize the virus and may also affect the inactivation efficiently by UV/TNP. The authors demonstrated this effect themselves in Figure 4, where more viral genome copies were detected in drier conditions, regardless of inactivation approach.

Response:

Agree. Previous experiments were performed by placing a 100ul aliquots of the viral suspension (in medium) as most of the inactivation assays were done in wet conditions. In this revised manuscript, we have included data of virus inactivation by placing a smaller droplet (10ul aliquot) of the viral suspension on a glass and TNP coated surfaces. Additionally, we dried these droplets (10ul) to mimic the natural deposition of the virus through respiratory droplets and drying quickly on surfaces. We evaluated viral inactivation on TNP surfaces when virus was dried or added as same size (10ul) liquid droplets and our data showed that TNP inactivates HCoV-NL63 present in either form. Importantly, our data also showed lower viral copies in dried samples on both TNP and control surface. This may confirm that drying itself can reduce infectivity. The details of virus drying and experimental set up is added in material and method section. The data is presented as Figure 4 in the revised manuscript.

 Which there is nothing inherently wrong with the authors’ approach in this manuscript, they must include the caveat in their Discussion (see also lines 345-346) that UV/TNP inactivation must also be analyzed in dried virus samples on various surfaces.

 Response:

We agree with your comment and a statement is added in the Discussion session. “It appears that dry environment has negative impact on coronavirus survival. Sizun et al have demonstrated that dried virus infectivity was for hours on aluminum, surgical gloves and sponges (doi: 10.1053/jhin.2000.0795). Also, the dried virus was shown to retain the viability for over 5 days on smooth surfaces at air-conditioned environments (https://doi.org/10.1155/2011/734690). The antiviral efficacy of the TiO2, therefore, could have a potential to prevent preserving the virus in the dry droplet as it will become inactivated before desiccation”.

 The authors have also not assessed virus inactivation in a biologically relevant medium, which is another important caveat to be addressed here.

Response:

In our experimental design we used virus suspended in supplemented culture medium, which contained 10% fetal bovine serum (FBS, Atlanta Biologicals), 2 mM L-glutamine, 25 U/mL penicillin, and 25 μg/mL streptomycin. We believe that the presence of FBS may provide some proteinaceous components to mimic the composition of the biological material present in respiratory and cough droplets. This information is included in the revised manuscript.

Section 3.3 Title – The authors find that increasing RH accelerates virus inactivation (Figure 4). However, the title of this section indicates that reduced RH enhances HCoV-NL63 inactivation, which is not consistent with their findings.

Response:

Thank you for the comment, we have modified the title to reflect the change: “Reducing the exposure distance and increasing the relative humidity enhanced HCoV-NL63 inactivation”.

Missing reference – Reference 20 is incomplete (missing access date).

Response:

We have provided a complete reference with access date.  

Line 19 – “to combat COIVD-19 is” should be “to combat COVID-19 are”.

Response:

We changed to ‘to combat COVID-19 are’. 

Line 146 – “was used for” should be “were used for”.

Response:

We changed to ‘were used for’. 

Line 253 – “amounts” should be singular.

Response:

Changed to ‘amount’ as suggested

Reviewer 2 Report

the comments are reported in the attached file

Author Response

We thank you for your time and effort in reviewing our manuscript and providing insightful comments. We have addressed all the concerns in this revised manuscript and point by point response is appended below.

The authors continue to favor and use molecular methods (qPCR) over infectivity tests in cell cultures which are the only ones able to truly establish whether a virus has been inactivated.

Response:

We fully understand your concerns but we have used both methods, RT-qPCR and infectivity assays, for testing the viral viability and our data shows comparable virus viability through both assays. Additionally, RT-qPCR approach has been commonly used for the detection of virus in the environmental samples (doi.org/10.1038/s41586-020-2271-3.; doi.org/10.1093/cid/ciaa905.; doi.org/10.3201/eid2607.200885.). Genomic approaches are shown to be highly sensitive in detecting viral RNA where multiple targets were used to generate a PCR amplicon (doi.org/10.1038/s41586-020-2271-3.; doi.org/10.1080/22221751.2020.1772678.). Thank you very much for your consideration.

Furthermore, as a demonstration of viral RNA fragmentation or not, following UV treatment, they use a PCR for a spike protein sequence that is only a relatively small part of the viral genome. But this sequence could be intact and the fragmentation could affect other parts of the viral genome. Therefore, also this approach is not methodologically correct.

Response:

As mentioned above, quantitative PCR data and viral infectivity assay for residual virus following treatment has shown comparable viral copies in both assays, therefore, we strongly believe that for the estimation of intact viral genomes, any small region of the viral genome can be utilized. We used a small region of the spike protein in our assay as a representative region. We totally agree that any fragmentation in any part of genome will inactivate the virus and detecting a small region ensures that virus has been inactivated beyond a doubt.

Other suggestions in particular:
Page 2, line 94: The authors did not clarify why they used two different cell lines, Vero E6 and HEK293L to cultivate the virus and for infectivity tests. If they have grown the virus in Vero cells, why then change the cell line?

Response:

We have included in the text that “HEK293L cells are highly permissive to HCoV-NL63 infection”, therefore were used for testing the infectivity. The virus stock was propagated in Vero cells as per the recommendation for BEI Resources. In our hands HEK293L cells are more susceptible to HCoV-NL63 than Vero cells but it might not be ideal to propagate the virus in HEK293L cells for subsequent passages. Thus, we followed BEI protocol for virus amplification and used HEK293L cells for viral infectivity assay.

 Page 3 line 99: HCoV-NL63 belongs to the family Coronaviridae, genus alphacoronavirus

Response:

Agree: changes are made in the manuscript to “HCoV-NL63 strain, is a human coronavirus and belongs to the family Coronaviridae, genus alphacoronavirus”.

Page 3 line 112: What does it mean: 100 microliter, Median Tissue Culture infectious dose (TCID50)? To this same question the authors replied: We used 100mL of the virus stock with known (calculated amounts) of the virus for inactivation assays. 100 microliters have become 100 mL, but of what? Please, is it possible to specify the number of TCID50?

Response:

We apologize for the typo, it was never a 100ml but 100ml, which has been corrected in the manuscript. We used HCoV-NL63 virus obtained from BEI Resources with 1.6x106 TCID50/ml (lot 70033870, NIAID, NIH) as our standard. We used 100ml of this virus (same lot every time) to extract RNA, synthesize cDNA, which was used for generating a standard curve. Using this standard curve, we calculated the number of viral copies in our stock propagated in Vero E6 cells in our laboratory. Once the viral copies were calculated we used that value to calculate the TCID which was 2.1x106/ml.  
Page 3. Virus infectivity assay: the true infectivity test is not described. Here the immunofluorescence test could be introduced.

Response:

Thank you for the suggestion: indeed, an infectivity is demonstrated by the detection of viral antigens, i.e. viral protein by either method, immunoblot, immunofluorescence or the detection of viral RNA through quantitative PCR. We detected viral antigen through immunofluorescence assay and positive sense RNA through RT-qPCR. The paragraph reads as

“The residual infectious viral copies (nonactivated copies) were determined by the localization viral protein of HCoV-NL63 through immunofluorescence assay, which relies on the production of viral antigens following replication/transcription. Therefore, detection of viral antigens in cells inoculated with UV-treated samples will confirm the presence of infectious virus and the lack of any viral antigen signal will reflect total viral inactivation. As a control of infection and replication, stock HCoV-NL63 was added onto HEK293L cells, stained for viral protein to demonstrate the functionality of this assay. Detection of signals in HCoV-NL63 infected cells but not in uninfected, control confirmed the specificity of detection (Fig. 3B, compare panel A with E). Our data showed that viruses from TNPs coated surfaces were completely inactivated even at 1 min exposure to UV-light, while surfaces without TNPs coatings had some infectious virus, demonstrated by the detection of immunofluorescent signal (Fig. 3B, compare panels B and F). Longer exposure to UV completely inactivated the virus demonstrated by the lack of immunofluorescent signals (Fig. 3B, panels C, D, G, H). These data confirmed that TNPs can enhance inactivation efficacies and even a brief exposure to the UV-light can effectively inactivate HCoV-NL63 virus rendering them non-infectious”

Page 4 line 166: cells (not slides) were permeabilized ...

Response:

Modified “cells” were permeabilized.

Page 6 Figure 3: the two small squares seem useless and also look bad

Response:

The intent is to show how the TNP coated slides look like, which may be important for the readers to know that TNO coated coverslips are slightly opaque, not transparent.

In Fig. 2, 3, 4 on the ordinate axis what is the unit of measurement? In the previous version it was the number of copies but here it has changed and it is not clear why.

Response:

The x-axis is still the same, which represent the number of viral copies (virus quant- through pPCR). We have changed the scale from logarithmic to linear scale. Logarithmic scale was not a good representation of actual viral copies, therefore, we changed it linear scale.
Figure 4 A can be eliminated. In the text it is reported that “it is practically not possible and unsafe to

expose surfaces with that high intensity of UV ...Therefore, we performed our assays at previously used distance ...” Therefore, why do these tests and report them in a special figure?

Response:

Thank you for the suggestion, we have removed “it is practically not possible and unsafe to

expose surfaces with that high intensity of UV. Therefore, we performed our assays at previously used distance” Now the paragraph reads as:

We also determined the effects of relative humidity on virus inactivation when the virus is deposited on TNP coated surfaces. Virus placed on TNP coated and uncoated coverslips were placed under three different relative humidity and exposed to the UV-light (76cm distance with 2900 µW/cm2) for indicated times. Our results showed that an increase in the relative humidity enhanced HCoV-NL63 inactivation with UV light on both control and TNP coated surfaces (Fig. 5B). Importantly, TNP coating enhanced the inactivation of HCoV-NL63 virus at all three tested relative humidity (Fig. 5B). This confirmed that surface inactivation of viruses can be achieved with shorter UV exposure on TNP coated surfaces and the humidity enhances the photocatalytic activity of TiO2.

References. Almost all of them must be checked. Sometimes there are no page numbers and any links to download the papers.

Response:

Thank you, we have checked all the references and have included page number and doi numbers to access them.

Round 2

Reviewer 2 Report

comments and suggestion are provided in the attached file

Author Response

This last version of the manuscript entitled “Human Coronavirus Inactivation by Titania Nanoparticle Coatings and UVC Radiation: Throwing Light on SARS-CoV-2” is enriched with interesting experimental data in the section 3.4. TiO2 coatings effectively inactivated HCoV-NL63 virus present in wet or dried form. Some minor corrections have been done with respect to the previous version. However, this new version retains some serious defects, basically the same of the previous one. Moreover, it shows a lot of new mistakes probably due to a hasty drafting. Below are my observations

Response:

We thank you for your time and effort in reviewing our manuscript. We conducted additional experiments including the infectivity assay to ensure that your comments are adequately addressed. We agree that PCR based assay should not used for detecting infectious virus but only for detecting the viral genomic RNA and detection of viral genome through RT-qPCR does not constitute the presence of infectious virus. We used virus infectivity assays whenever needed along with the direct detection of viral genome through RT-qPCR and both assays corroborate very well.

All these changes are incorporated in this revised manuscript.

 Titer: “In vitro inactivation” is a useless change, a bit pretentious “Throwing Light on SARS-CoV2”

Response:

When we submitted this article to biorxiv, the screening editor ensured that we specified our method is for in-vitro inactivation of virus not inside living beings, that’s why ‘In-vitro’ was added but have removed it in this revised version as per your suggestion.

Page 3, 2.2. It is not clear why the method used to cultivate in Vero cells the virus received from BEI resources is described since in subsequent experiments the original virus received from BEI will be used and not the one cultivated by the authors.

Response:

We apologize for the confusion, we cultivated HCoV-NL63 using stock from BEI (1.6x106 TCID50/ml) in Vero Cells. The collected supernatant from these cells were stored at -80oC until use. TCID50/ml of this in-house virus stock was determined by infecting the monolayer of Vero E6 cells, which was determined to be 2.1x106TCID50/ml. Therefore, we included virus cultivation in Vero E6 cells.

Line 104-108: “and virus was quantified by Reverse Transcriptase-quantitative PCR (RT-qPCR). Harvested virus TICD was determined using the HCoV-NL63 (1.6x106 TCID50/ml; lot# 70033870, NIAID, NIH) containing 320 copies per 100 μl (where does this data come from? is this information declared by NIAID, NIH?). The standard curve was generated using the HCoV-NL63 virus (lot #70033870) and was applied to determine the TICD of the harvested virus which was 2.1 x106/ml. An aliquot (100 μl) containing HCoV-NL63 of 2.1x105 TCID50.” this whole paragraph is full of repetitions and errors. Line 104: TCID instead of TICD.

line 108-109: An aliquot (100 μl) containing HCoV-NL63 of 2.1x105 TCID50: this sentence must be completed or deleted.

Response:

We have simplified this paragraph and removed unnecessary content including repetitions and have completed the sentence. Now the paragraph reads as:

 HCoV-NL63 strain, is a human coronavirus and belongs to the family Coronaviridae, genus alphacoronavirus. HCoV-NL63 was obtained from BEI Resources (1.6x106 TCID50/ml; lot# 70033870, NIAID, NIH) and propagated in Vero cells by infecting the Vero cell monolayer with HCoV-NL63 for 2 h. Unattached virus was removed by washing followed by addition of fresh medium. Supernatant containing virus was harvested after 7 days, cell debris were removed by centrifugation before storing at -80oC. TCID50 of the harvested virus was determined indirectly by RT-qPCR using a standard curve generated as described later (2.6. RNA extraction and qPCR). By this method the harvested virus showed to have 2.1x106TCID50/ml. An aliquot (100 µl) of this supernatant (HCoV-NL63) was used for extracting RNA and Reverse Transcriptase qPCR (RT-qPCR) for quantifying viral copies. All the assays were conducted under biosafety level 2+ (BSL-2+) containment.

 Calculating the TCID 50 by PCR is quite anomalous. It seems an attempt to get around the obstacle of calculating this figure with a virological method more time consuming and more challenging but more suitable. On the other hand, this tendency characterizes most of this manuscript. Anyhow, more simply it could be written (if that's the meaning) “Harvested virus TCID50 were determined indirectly by RT-qPCR using a standard curve generated as described later (2.6 RNA extraction and qPCR). By this method, the harvested virus showed to have 2.1x106TCID50/ml. “

Response:

Agree and have modified the sentence as follows:

“TCID50 of the harvested virus was determined indirectly by RT-qPCR using a standard curve generated as described later (2.6. RNA extraction and qPCR). By this method the harvested virus showed to have 2.1x106TCID50/ml. An aliquot (100 µl) of this supernatant (HCoV-NL63) was used for extracting RNA and Reverse Transcriptase qPCR (RT-qPCR) for quantifying viral copies. All the assays were conducted under biosafety level 2+ (BSL-2+) containment”.

Page 3 line 117: An aliquot of HCoV-NL63 virus (100 μl, Median Tissue Culture Infectious dose (TCID50) … This sentence as previously reported is not understandable. It can be changed as: An aliquot (100 μl, containing 2.1x105 TCID50) of HCoV-NL63 virus ….

Response:

Agree and have modified the sentence as follows:

An aliquot (100 μl, 2.1x105 TCID50) of HCoV-NL63 virus was placed on TNPs-coated and uncoated coverslips (18 mm diameter, 1017.88 mm2) and exposed to the USHIO Germicidal Lamp (model G30T8), which generates UV-C light (wavelength: 254nm, 99V, 30W, 0.355A) for various time points”

Page 3 line 137: Similar amounts of liquid droplets on virus on Dried virus containing coverslips were exposed to UV…even this sentence is not understandable.

Response:

Agree and have modified the sentence as follows:

“Same volume of liquid droplets, from the same virus stock, were deposited onto coverslips or TNPs-coated surfaces and exposed to the UV light for 30 sec or 1 min without drying the droplets”.

Page 3 line 138-139: Dried virus on either TNPs-coated or uncoated coverslips were exposed to the UV light without drying. ??? it is not clear, the virus was or was not dried?

Response:

We apologize for the repetitions. We have fixed those sentences and now read as follows”

“For drying the liquid droplets, ten small aliquots (10 μl each) for a total of 100 μl, were deposited on either TNPs-coated or uncoated coverslips and allowed to visibly dry (20 min, room temperature, 45% humidity) before exposing to the UV light. Same volume of liquid droplets, from the same virus stock, were deposited onto coverslips or TNPs-coated surfaces and exposed to the UV light for 30 sec or 1 min without drying the droplets.”

Page 3 lines 142-143. “…dissolved content was collected and applied onto a permissive, human embryonic kidney (HEK293L) cell monolayer and estimating viral copies …” why wasn't the immunofluorescence test done? Moreover, the choice of this cell line instead of the Vero was not clarified by the authors' response. They speak of a greater sensitivity of these cells which, however, is not documented either by the authors or by the literature.

Response:

Immunofluorescence assays was done on the cells infected with residual and control virus, we have included that information and the sentence reads as’

For determining the residual infectious virus, dissolved content was collected and applied onto a permissive, human embryonic kidney (HEK293L) cell monolayer for the infectivity assay. To ensure that all the residual virus was recovered from the control and TNP coated surfaces, 100 μl PBS was added and incubated at 37oC for 30min (3x) and combined before adding onto the target cells.

Regarding infectivity of HEK293L cells:  Study conducted by Kaye et al have demonstrated that NL63 coronavirus could be detected on day 4 in both, HEK293L and Vero cells. However, virus remained detectable on days 7 and 11 only in HEK293 cells, while it was not detected in Vero cells (https://www.ncbi.nlm.nih.gov/pmc/articles/PMC3291385/). In another study, virus titer in HEK293 was demonstrated as twice as high as compared to that in Vero cells (https://www.ncbi.nlm.nih.gov/pmc/articles/PMC3716658/). Therefore, we concluded that HEK293L cells would be better suited for detecting even a low numbers of virus particles still present in the samples after UV exposure.

Additionally, we conducted an infectivity assay of HCoV-NL63 in both Vero and HEK293L cells and localized viral protein through immunofluorescence assay. Our results showed that HEK293L cells were equally permissive if not better to HCoV-NL63 infection. This data has been included as Figure 2C. Vero or 293L cells were infected the with same amount of HCoV-NL63, incubated for 48h followed by immune localization of viral proteins. Using polyclonal rabbit anti-CoV antibody (1:200; BEI Resources, NIAID, NIH) for 1h at RT, followed by incubation with goat anti-rabbit Alexa Fluor 488 (green signals). Nuclei was stained with TO-PRO-23 (blue). A representative image shows infection of HEK293L cells with HCoV-NL63.

Page 4. In the previous review I had recommended: ”Virus infectivity assay: the true infectivity test is not described. Here the immunofluorescence test must be introduced.” I repeat what was previously recommended: the true infectivity test, in this paper the immunofluorescence assay, must be entered here. RT-qPCR is not an infectivity test! This is an absolutely wrong concept and the authors must realize it. Only very indirectly it can provide data also useful for establishing the infectivity of a virus.

Response:

We agree and have introduced virus infectivity assay through immunofluorescence. We also detected virus copies in the infected cells by extracting total RNA and RT-qPCR. Detection of viral genomic RNA in the cells infected with inactivated virus will be from the infection and replication of the virus present in the material added onto the cells.

2.5. Virus Infectivity and immunofluorescence assay (IFA))

Control or the UV-treated viruses were added onto the human embryonic kidney cells for 2 h (370C, 5% CO2) before washing the dead and disintegrated virus and adding fresh medium. These cells were incubated for 48h at 370C in a humidified chamber supplemented with 5% CO2. Since an infectious virus enters the cells to replicate and produce viral genomic RNA and proteins, detection of intracellular viral RNA and viral proteins, confirms the presence of live virus. Infected cells were harvested for the extraction of total RNA and the detection of intracellular viral genomic RNA through RT-qPCR (2.6. RNA extractions and qPCR). Viral proteins were detected through immunofluorescence assay in which infected cells were fixed using 3:1, methanol: acetone and stored at -800C until used. Cells were permeabilized with 0.1% Triton X-100 for 30 min, washed (3x) and blocked (3% normal donkey serum, 0.5% BSA) for 1h at RT. Cell monolayers were washed again (3x) and incubated with polyclonal rabbit anti-CoV antibody (1:200; BEI Resources, NIAID, NIH) for 1h at RT, followed by incubation with goat anti-rabbit Alexa Fluor 488 (1:5000; Molecular Probe, Carlsbad, CA), secondary antibody for 1h at RT in the dark. Finally, the nuclei were stained with TO PRO-3 (ThermoFisher, Walthman, MA). Coverslips were mounted on the glass slides using antifade and the slides were examined using Carl Zeiss LSM 780 confocal laser scanning microscope. 

Page 4 line 144-146: “To ensure the efficacy of virions recovery from control and TNP coated surfaces, cover slips were incubated with PBS at 37oC for 30min (3x) and combined before using for infection assays”. It had already been said that cover slips were incubated with PBS at 37oC for 30min (3x). But “combined” what does it mean?

Response:

We apologize for the repetitions. We have fixed those sentences and now read as follows”

“To ensure that all the residual virus was recovered from the control and TNP coated surfaces, 100 μl PBS was added and incubated at 37oC for 30min for three subsequent times followed by combining them before adding onto the target cells.”

Page 5 line 184-185: rabbit anti-CoV antibody. Against which viral protein is this antibody directed to?

Response:

This was a polyclonal antibody for detecting spike protein. We added this information in the IFA description: Cell monolayers were washed again (3x) and incubated with polyclonal rabbit anti-CoV spike antibody (1:200; BEI Resources, NIAID, NIH) for 1h at RT, followed by incubation with goat anti-rabbit Alexa Fluor 488 (1:5000; Molecular Probe, Carlsbad, CA), secondary antibody for 1h at RT in the dark.

Page 5. “Presence of the viral RNA and its integrity was determined using qPCR approach as it was previously used for detection of SARS-CoV-2 in environmental samples [13,22-23]. Genomic approaches are shown to be highly sensitive in detecting viral RNA and multiple targets are used for generating PCR amplicon [13,24]. Any region of the viral genome can be used for detecting genome integrity and for our assays, we selected S protein coding gene as this protein is essential for binding to the receptor and fusion with the host cell membrane [25]. Since fragmentation of any region of the viral genome would render the virus non-infectious, we used spike region as a test of viral genome integrity.” Some of these claims are fairly obvious, others are questionable. The sensitivity of PCR for viral nucleic acid detection is beyond question, it is well known for many years and therefore the phrase is obvious. The reason given for selecting the gene for protein S as the target of RT-qPCR does not make much sense in this context. A more valid reason could be the greater sensitivity of this PCR compared to other PCRs with other targets. The true question here, once again, is whether PCR nucleic acid testing is suitable for determining the infectivity of a virus. At this purpose I report extracts from two of a large amount of literature references. The first is cited by the authors themselves:

  1. Aerosol and Surface Distribution of Severe Acute Respiratory Syndrome Coronavirus 2 in Hospital Wards, Wuhan, China, 2020. “We tested air and surface samples for the open reading frame (ORF) 1ab and nucleoprotein (N) genes of SARS-CoV-2 by quantitative realtime PCR. Our study has 2 limitations. First, the results of the nucleic acid test do not indicate the amount of viable virus. “
  2. APPLIED AND ENVIRONMENTAL MICROBIOLOGY, Dec. 2006, p. 7671–7677: Besides, from our results, detection of the fragment of the genome was not reliable for the infectivity and may not yet be the critical parameter to determine the risk of viruses from water treated with UV

Response:

We apologize for any confusion, we have modified significantly and our intent is to not claim that RT-qPCR can be used for infectivity assay. RT-qPCR assay is just for detecting the intactness of the viral genome. We have conducted infectivity assay and immunofluorescence assays to detect live virus. These data have been included.

Additionally, we have removed the reason, as stated earlier, it is indeed used because of high sensitivity and specificity. The entire paragraphs read as follows:

“Importantly, detection of genomic RNA through RT-qPCR does not constitute for the presence of in ‘infectious’ virus but gives an indication for the presence of viral genomic RNA. Therefore, a virus infectivity assay must be performed for the detection of any infectious virus. However, if there isn’t detection of any intact viral genomic RNA, in samples after treatment/exposure, it would constitute that virus has been completely eliminated. Any region of the viral genome can be used for detecting genome integrity and for our assays, we selected Spike protein, which helps in attaching the virus to host cells [25], because of its high sensitivity and specificity for the detection of HCoV-NL63. HCoV-NL63 viral copies were calculated using standard curve generated on the basis of serial dilutions of known HCoV-NL63 genomic RNA. Expectedly, the copies of intact viral genomic RNA declined with UV-exposure (Fig. 2A). Importantly, we saw an efficient reduction in the number of intact viral genomic RNA even at 1 min exposure to the UV-light. However, there were some intact genomic RNA of HCoV-NL63, at least in the region (spike protein) used for detection in our qPCR assay but increasing the exposure time to 5, 10 and 30 min, completely degraded the genomic RNA of HCoV-NL63, below the detection limit, confirming total disintegration of the viruses with UV (Fig. 2A). Since, RT-qPCR determines the intactness of viral genomic RNA, we asked whether treatment with UV can reduce the infectivity of HCoV-NL63 virus. To this end, UV exposed HCoV-NL63 viruses were added onto the monolayer of HEK293L for 2 h to facilitate attachment and entry of the residual live virus. The permissive nature of HEK293L cells were evaluated, before using them for the estimation of residual live virus, by adding same amounts of HCoV-NL63 on Vero E6 or HEK293L cells and allowing them to grow for 48h after removing the unattached virus. Infection and replication of HCoV-NL63 virus in these cells were detected by localizing viral spike glycoprotein through Immunofluorescence assay”.   

Page 5 lines 212-213 “However, there were still some intact copies of HCoV-NL63, at least in the region (spike protein) used for detection in our qPCR assay”. Therefore, the authors recognize that the viral RNA could be intact in the amplified tract but not in other segments. Accordingly, the RNA could be degraded even if the PCR used was positive and thus the virus could be inactivated.

Response:

We have clarified that degradation in any region of the viral genomic RNA will inactivate the virus and longer exposure totally disintegrates the virus. The sentence reads as follows:

“However, there were some intact genomic RNA of HCoV-NL63, at least in the region (spike protein), but other regions may have been degraded RNA to make the virus inactivated, but increasing the exposure time to 5, 10 and 30 min, completely degraded the genomic RNA of HCoV-NL63, below the detection limit, confirming total disintegration of the viruses with UV.”

Page 5 line 215-216: “Since, the RT-qPCR determines the intactness of genomic RNA, we further determined whether the UV-treatment reduced the infectivity of HCoV-NL63 virus.” sentence not very clear. Its meaning seems to me: we tried to verify if the UV treatment in addition to decreasing the level of viral RNA caused a decrease in the infecting virus.” Good! This is correct, a true infectivity assay in cell cultures. Unfortunately, instead of using the immunofluorescence test described in the methods, the authors employ a strange and questionable molecular approach.

Response:

We performed infectivity assay and have detected the viral protein through immunofluorescence assay where the virus was exposed to UV either on TNP or control non-coated surface. Immunofluorescence is primarily a qualitative assay, therefore, we also tried detecting the copies of the viral genome following replication after entering into the cells. We have modified the sentence as follows: 

Since, RT-qPCR determines the intactness of viral genomic RNA, we asked whether treatment with UV can reduce the infectivity of HCoV-NL63 virus. To this end, UV exposed HCoV-NL63 viruses were added onto the monolayer of HEK293L for 2 h to facilitate attachment and entry of the residual live virus.

Page 6 line 239-240: “the viral copies quantified through RT-qPCR for viral genome fragmentation” The RT-qPCR is for viral genome detection and quantification.

Response:

We have modified the sentence, which reads as follows: 

“Expectedly, the viral copies quantified through RT-qPCR for viral genome fragmentation showed significantly reduced copies (almost to the background level) of HCoV-NL63 from the TNP coated surface as compared to the control (uncoated surface) even at 1-min of UV-exposure (Fig. 3A).”

Page 7, line 260 “To correlate virus inactivation with viral infectivity” it is not clear what the authors want to correlate.

Response:

We have modified the sentence, which reads as follows: 

“To demonstrate whether the viral genomic RNA detected after the UV-treatment on the TNP or control surface had any infectious virus, a virus infectivity assays was performed by adding the recovered virus onto the monolayer of HEK293L cells”.

Line 274-275: “even a brief exposure to the UV-light can effectively inactivate HCoV-NL63 virus rendering them non-infectious.” If a virus is inactivated it is not infectious, there are no inactivated viruses that are capable of infecting target cells.

Response:

We have modified the sentence, which reads as follows: 

However, longer exposure to UV inactivated the virus to undetectable levels, demonstrated by the lack of immunofluorescent signals (Fig. 3B, panels C, D, G, H). These data confirmed that virus on the TNPs coated surface can be inactivated quickly as compared to the non-coated control surface.

In all Figures in the charts the indication “virus quant” is not correct, it should be replaced with number of virus copies.

Response:

Virus quant has been changed to ‘Number of virus copies (RT-qPCR).

In the section 3.2 finally a virological method was employed, the immunofluorescence assay! Unfortunately, this method was immediately abandoned to return exclusively to Rt-qPCR with all the limitations already discussed.

Response:

We used immunofluorescence assay for detecting the residual infectious virus following treatment, which was in agreement with RT-qPCR data, therefore, we used this assay for determining the residual copies of the HCoV-NL63. We also believe that qPCR data can provide quantitative results while IFA is more qualitative.

Page 9 line 320: Figure 4. Discussion. Delete Figure

Response:

We think that Figure 4 provides important information and thus should be retained.

Page 10, lines 381-384 the legend of Fig 6 is deleted and the same text is written at page 11 line 394-397. Why???

Response:

We apologize for the reshuffling of the text, which we have fixed in this revised version. We believe that this may happened during uploading of the manuscript.

References: the link of ref. 42(Buonanno et al …) is wrong Ref 47 , 48 and 57 are in part deleted. ???

Response:

We apologize for incomplete references, which are corrected in this revised manuscript.

Ref. 42- Buonanno, M.; Welch, D.; Shuryak, I.; Brenner, D.J. Far-UVC light (222 nm) efficiently and safely inactivates airborne human coronaviruses. Scientific Reports 2020, 10, 1-8, 2020, 10, 1-8, https://doi.org/10.1038/s41598-020-67211-2

Regarding references 47, 48 and 57, we ensured for their accuracy by clicking on the doi, which opens up the respective intended article.

In conclusion What does this work show and what should therefore be highlighted?

Response:

We have included a concluding summary as suggested.

In conclusion, the photoactive TNPs have strong virucidal effect, which are preserved under different environmental condition. In most cases, the genomic copies of human coronavirus NL63 from TNP coated surface was undetectable within 1 min of UV exposure, which was also confirmed by cell infectivity assay. In an in-vitro virus inactivation assay, when the detection of the viral RNA by the RT-qPCR is negative (or below the detection limit) it can be concluded that viral genome has been completely disintegrated. However, in case of a positive RT-qPCR results, viral genome in the area of PCR amplification will be intact but fragmentation in other region of the viral genome as well as inactivation of surface glycoprotein will make virus non-infectious. Therefore, a virus infectivity assay must be performed to determine the levels of infectious virus. Thus, lack of any RT-qPCR signals can be construed as total inactivation of the virus though disintegration of viral genomic RNA.
